# Identifying Candidate Genes for Hypoxia Adaptation of Tibet Chicken Embryos by Selection Signature Analyses and RNA Sequencing

**DOI:** 10.3390/genes11070823

**Published:** 2020-07-20

**Authors:** Xiayi Liu, Xiaochen Wang, Jing Liu, Xiangyu Wang, Haigang Bao

**Affiliations:** 1National Engineering Laboratory for Animal Breeding, Beijing Key Laboratory of Animal Genetic Improvement, College of Animal Science and Technology, China Agricultural University, Beijing 100193, China; liuxiayi@cau.edu.cn (X.L.); jing921580312@163.com (J.L.); 2Chinese Academy of Sciences Key Laboratory of Animal Ecology and Conservation Biology, Institute of Zoology, Beijing 100101, China; 15600911531@163.com; 3College of Life Sciences, University of Chinese Academy of Sciences, Beijing 100049, China; 4Key Laboratory of Animal Genetics, Breeding and Reproduction of Ministry of Agriculture and Rural Affairs, Institute of Animal Science, Chinese Academy of Agricultural Sciences, Beijing 100193, China

**Keywords:** Tibet chicken, hypoxia adaptation, signatures of selection, RNA sequencing

## Abstract

The Tibet chicken (*Gallus gallus*) lives on the Qinghai–Tibet Plateau and adapts to the hypoxic environment very well. The objectives of this study was to obtain candidate genes associated with hypoxia adaptation in the Tibet chicken embryos. In the present study, we used the fixation index (Fst) and cross population extended haplotype homozygosity (XPEHH) statistical methods to detect signatures of positive selection of the Tibet chicken, and analyzed the RNA sequencing data from the embryonic liver and heart with HISAT, StringTie and Ballgown for differentially expressed genes between the Tibet chicken and White leghorn (*Gallus gallus*, a kind of lowland chicken) embryos hatched under hypoxia condition. Genes which were screened out by both selection signature analysis and RNA sequencing analysis could be regarded as candidate genes for hypoxia adaptation of chicken embryos. We screened out 1772 genes by XPEHH and 601 genes by Fst, and obtained 384 and 353 differentially expressed genes in embryonic liver and heart, respectively. Among these genes, 89 genes were considered as candidate genes for hypoxia adaptation in chicken embryos. ARNT, AHR, GSTK1 and FGFR1 could be considered the most important candidate genes. Our findings provide references to elucidate the molecular mechanism of hypoxia adaptation in Tibet chicken embryos.

## 1. Introduction

Hypoxia was found to be the most important stimulating factor for chicken embryonic development in chicken embryos hatched in high altitude areas [1,2,3,4]. It reduced chicken hatchability [1,5] and retarded the growth of chickens after incubation [6]. When the partial pressure of oxygen in the incubator was maintained at the same level as the lowland level, the hatchability in the high-altitude area was similar to that in the lowland area [1]. The Tibet chicken (*Gallus gallus*, a domestic species) originates on the Qinghai–Tibet Plateau and mainly lives over the agri-nomadic area with the altitude of 2200–4100 m in the Southwest of China [7]. It adapted itself quite well to hypoxia and showed higher embryo hatchability in comparison with lowland chicken breeds under hypoxia [1,7,8,9]. The White leghorn (*Gallus gallus*), Shouguang chicken (*Gallus gallus*) and Silkie chicken (*Gallus gallus*) are all lowland domestic chicken breeds and originate from lowland areas with no more than 1000 m altitude. They do not adapt to high altitude hypoxia environments. In the lowland area (50 m, Beijing), the Tibet chicken and lowland chickens show similar hatchabilities [1,7]. Under the low oxygen conditions of 15% (equivalent to the partial pressure of O_2_ at a 2900 m altitude in Linzhi, China), the hatchability of the Tibet chicken embryo could still reach about 80% and the hatching rate of lowland chickens was usually less than 40% [1,7]. Under the low oxygen conditions of 13% (3650 m, Lhasa, China), the hatchability of the Tibet chicken was about 30%, whereas the hatchabilities of lowland chicken breeds were less than 10% [7].

Signatures of positive selection are chromosome regions that contain advantageous mutations related to enhanced fitness or productive ability [10,11]. Those regions with hard selective sweeps were more easily detected than those with soft selective sweeps, and display several characteristics as follows [11,12,13,14]: (1) increasing or fixed frequencies of favorable alleles for animal phenotype selected by nature or farmers [11]; (2) lower genetic polymorphism and long-range linkage disequilibrium (LD) or haplotypes than unselected chromosome regions because of selective sweep [12] or hitchhiking effect [14,15] in the selected populations; (3) larger allele frequency differences between the selected and unselected populations compared with neutral evolution regions [13]. Soft selective sweeps are more common and also more difficult to detect than hard sweeps because of their weaker effects on linked sites [16,17]. Some confounding factors, such as purifying and background selection, demography and migration, can affect the results of positive selection analysis [16]. Many statistical methods were developed to search for signatures of positive selection based on these characteristics above, such as Tajima’s D [18], cross population extended haplotype homozygosity (XPEHH; [19]), integrated extended haplotype homozygosity (iHH; [19]), pooled heterozygosity (Hp; [20]), fixation index (Fst) [21] and so on. Candidate genes of some traits of domestic animals were detected with different selection signature methods [10,12,22,23,24,25]. These methods above were also used to detect candidate genes for high altitude hypoxia adaptation in plateau animals and some candidate genes for hypoxia adaptation had been detected, such as *EPAS1*, *EGLN1*, *CTGF* and *AMOT*, etc. [9,26,27,28,29,30,31]. Using the Hp method and RNA sequencing analysis, Zhang et al. [9] identified only one believable candidate gene (*CTGF*) for chicken embryonic hypoxia adaptation, which was not an ideal result for the quantitative trait. Different statistical methods are biased for different types of signatures of selection, for example, XPEHH may be more suitable to detect selected genes which are fixed or close to be fixed in one population but remain polymorphic in the whole population [19]. Therefore, although Zhang et al. [9] found one believable candidate gene, we believe that it is still necessary to use different selection signal methods and data to detect candidate genes for hypoxia adaptation in chicken embryos.

In the present study, a total of 71 Tibet chickens and 113 lowland chickens were genotyped with 600 K SNP Axiom^®^ Genome-Wide Chicken Genotyping Array [32] (Affymetrix, Inc. Santa Clara, CA, USA), and the Fst and XPEHH test methods were used to detect signatures of positive selection in the Tibet chicken. Differentially expressed genes (DEGs) were checked with RNA sequencing data from embryonic livers and hearts under hypoxia conditions. Genes which were screened out by both selection signature analysis and RNA sequencing analysis could be regarded as believable candidate genes for hypoxia adaptation of chicken embryos, and 89 believable candidate genes for embryonic hypoxia adaptation were screened out. This work provides a reference for further studies to explain the molecular mechanism of hypoxia adaptation in Tibet chicken embryos in the future.

## 2. Materials and Methods

### 2.1. Ethics Statement

All experimental procedures and animals used were approved by the Ethics Review Committee for Laboratory Animal Welfare and Animal Experiment of China Agricultural University (Approval number: AW09060202–1).

### 2.2. DNA Samples and Single Nucleotide Polymorphism (SNP) Genotyping

Seventy-one of the Tibet chicken blood samples were collected from 3 provinces of China (23 samples from Beijing, 43 samples from Tibet and 5 samples from Yunnan). One hundred and thirteen lowland chicken blood samples, including 40 samples of White leghorn, 43 samples of Silkie chicken and 30 samples of Shouguang chicken, were collected from the Experimental Chicken Farm of China Agricultural University. DNA extraction from chicken blood was performed using the TIANamp Blood DNA Kit (Cat. DP348, TIANGEN) according to the protocols supplied. All DNA samples were genotyped with 600K SNP Axiom^®^ Genome-Wide Chicken Genotyping Array ([32], Affymetrix, Inc. Santa Clara, CA, USA).

### 2.3. RNA Samples and RNA Sequencing Data Collection

Fertilized eggs of the Tibet chicken and White leghorn were collected and incubated in a simulated hypoxic incubator with 13% oxygen concentration. At the end of day 16 of incubation, eggshells were opened at the air-cell and living embryos were pulled out with a nipper, then their hearts and livers were separated out and immediately stored in liquid nitrogen for RNA extraction. A total of eight samples were collected at random, and each kind of sample has two biological replicates.

Total RNA was isolated using TRIzol™ Reagent (Life Technologies Invitrogen, Carlsbad, CA, USA) according to the manufacturer′s instructions. Complementary DNA (cDNA) library construction was performed using TruSeq™ RNA Library Prep Kit (Illumia, San Diego, CA, USA) according to the protocols supplied. After purification of index-coded PCR products, samples were clustered with TruSeq PE Cluster Kit v3-cBot-HS (Illumia, San Diego, CA, USA) strictly according to the protocols supplied. After generating clusters, paired-end sequencing (PE, 100 bp) was performed using the Illumina HiSeq™ 2000 platform by commercial service companies. After the adapter sequences in raw data were trimmed, reads with more than 10% of undetermined bases (N) were removed. Clean reads were obtained after removing low-quality bases (Qphred < 20) and then aligned against the chicken genome reference (GRCg6a) with HISAT2 [33].

Four RNA sequencing data of hearts of Tibet chicken and White leghorn embryos (GEO accession No: GSE77166) were downloaded and analyzed together with our own data, as those heart samples were subjected to the same experimental treatment process as the heart samples we collected [9].

### 2.4. Identifying Signatures of Positive Selection by XPEHH Test and Fst Statistics

Data of SNP genotyping were controlled using PLINK software (Version 1.90b3.34; [34]) with parameters as follows: geno 0.1, hwe 1 × 10^−6^, maf 0.01, mind 0.1, chr 1–28. BEAGLE (Version 3.3; [35]) analyses were performed to impute missing SNP data and SELSCAN (version 1.1.0b; [36]) was used to calculate XPEHH scores. The XPEHH scores were standardized with means and variances and the threshold of XPEHH scores (corresponding to *p*-value < 0.01) was estimated with the standard normal distribution [37]. The chromosome regions of signatures of selection were determined by clustering the significant core SNPs (*p*-value < 0.01) of which the distance from their nearby significant core SNPs was less than 200 kb [37]. The fixation index (Fst) [21] was calculated with VCFtools (Version 0.1.13; [38]) with a window size of 30 K and a window-step of 15 K. Windows with less than 5 SNPs were discarded. Windows with the first 2% of the Fst value of each chromosome were considered candidate regions of signatures of selection. 

The two statistical methods have different biases to the signatures of selection, so all the signatures of selection screened by these two methods are considered to be candidate selected regions [19,21].

### 2.5. Screening for Differentially Expressed Transcripts and Genes 

The expression level for each gene was estimated with the fragments per kilobase of transcript per million fragments mapped (FPKM) value [39]. Differentially expressed genes (DEGs) were analyzed following the workflow with HISAT2, StringTie and Ballgown [40]. Briefly, clean data were aligned against the chicken reference genome (GRCg6a) with HISAT2 (Version 2.1.0; [33]). The SAM (Sequence Alignment/Map) files were sorted and converted to BAM (compressed Binary version of the sequence Alignment/Map format) files with SAMtools (https://github.com/samtools/samtools/releases/tag/1.9, Version 1.9). Transcripts of each sample were assembled with StringTie (Version 2.0; [41]) based on Gallus_gallus.GRCg6a.97.gtf (ftp://ftp.ensembl.org/pub/release-97/gtf/gallus_gallus/) and each sample’s BAM file. After assemblies, samples’ GTF (General Transfer Format) documents and Gallus_gallus.GRCg6a.97.gtf were merged together with the StringTie "-merge option and transcript abundances of each sample were estimated with the StringTie -eB option [40]. Transcripts with a variance across samples less than one were removed and then differentially expressed transcripts and genes between embryos of the Tibet chicken and White leghorn were screened using the stattest function from Ballgown (Version 2.12.0; [42]) with the getFC = TRUE parameter. The batch effect of two sources of transcriptome data were considered during our analysis. Transcripts and genes with a fold change >3.0 and *p* value < 0.05 between the Tibet chicken and the lowland chicken were identified as differentially expressed transcripts and genes. All DEGs, including the genes corresponding to the differentially expressed transcripts and differentially expressed genes screened out by Ballgown, were used for subsequent steps.

### 2.6. Identifying Candidate Genes for Hypoxia Adaptation of the Tibet Chicken Embryos

All genes which were detected by Fst and/or XPEHH were considered candidate genes for the Tibet chicken to the high altitude environment in Qinghai–Tibet Plateau, but only those genes that were detected to be differentially expressed by the transcriptome analyses as well as being screened by the selection signature tests were considered to be believable candidate genes for hypoxia adaptation of the Tibet chicken embryos.

### 2.7. Functional Annotation of Candidate Genes for Hypoxia Adaptation of the Tibet Chicken Embryos

Functional annotation of differentially expressed genes from different tissues were performed with DAVID 6.8 (https://david.ncifcrf.gov/) by default, respectively.

## 3. Results and Discussions

### 3.1. Analyses for Signatures of Selection

The 600 K SNP Axiom^®^ Genome-Wide Chicken Genotyping Arrays used in this study contained a total of 580,961 SNPs. After quality control, only 477,838 SNPs from chromosome 1–28, LGE22C19W28_E50C23 and LGE64 of the chicken genome were used to screen signatures of selection. All samples of 71 Tibetan chickens and 113 lowland chickens passed the quality control by PLINK software (Version 1.90b3.34; [34]). The Manhattan plot of XPEHH scores is plotted in Figure 1 and the Manhattan plot of Fst statistics is shown in Figure 2. A total of 1772 candidate genes were identified by XPEHH test (Appendix A; Figure 3A) and 601 candidate genes were screened by an Fst test (Appendix A; Figure 3A). There were only 86 overlapping genes between the results of the two tests (Appendix A; Figure 3A).

The XPEHH test is a powerful method used to detect selected alleles which have risen to a high frequency or fixation within a short period based on the long-range haplotype without erosion by recombination [19]. The fixation index (Fst), a measure of population genetic differentiation, was calculated to identify signatures of selection based on the divergence of allele frequencies among populations [43,44]. Although the Fst method is also sensitive in detecting fixed selection signatures [10], the test power of XPEHH is significantly higher than that of the Fst method under the same parameters of marker interval distance, frequency of the selected allele, sample size or selection coefficient [44]. The correlation between the results of the two methods was very low (0.03) in chicken data [44]. In the present study, less than 5% of the 1771 genes were also detected by Fst, and most of the genes detected by Fst (85.6%) were not detected by XPEHH, which may indicate that the two methods have different priorities in detecting the selected genes. The overlapping genes may indeed be related to positive selection signatures, but we cannot rule out others. Therefore, all genes detected by the two methods were combined to overlap with differentially expressed genes from RNA sequencing analyses to get believable candidate genes in the present study (Figure 3).

### 3.2. Analysis for RNA Sequencing Data

A total of 384 differentially expressed known genes (DEKGs; Appendix A; Figure 3B) were found between the embryonic livers of the Tibet chicken and the lowland chicken. These genes may play important roles in hypoxia adaptation of embryonic livers. To understand the role of these genes in hypoxia adaptation, the Gene Ontology (GO) and Kyoto Encyclopedia of Genes and Genomes (KEGG) enrichments were performed and the results are shown in Figure 4. From Figure 4, we could find that many terms were obviously enriched relating to fatty acids (gga00071: fatty acid degradation; gga04146: peroxisome; GO:0000062 ~ fatty-acyl-CoA binding), energy production (gga00020: citrate cycle (TCA cycle); gga00010: glycolysis/gluconeogenesis; gga01200: carbon metabolism; gga01210: 2-oxocarboxylic acid metabolism; gga00310: lysine degradation, GO:0006099 ~ tricarboxylic acid cycle; GO:0005739 ~ mitochondrion) and antioxidant stress (gga04146: peroxisome; gga00480: glutathione metabolism; GO:0006749 ~ glutathione metabolic process; GO:0043619 ~ regulation of transcription from RNA polymerase II promoter in response to oxidative stress; GO:0004364 ~ glutathione transferase activity). These results indicated that the adjustment of fatty acid metabolism, energy metabolism and antioxidant stress processes in the liver may be important aspects of chick embryos achieving hypoxic adaptation. These results were consistent with previous studies in different animals [45,46,47,48,49,50]. Hypoxia-inducible factor 1 (HIF-1) is one of the key transcription factors for animal hypoxia adaptation [51,52]. Elevated fatty acid metabolism could decrease the stabilization of the alpha subunit of HIF-1 which was vital for animal survival in hypoxia [46]. Skeletal muscle decreased fatty acid oxidation in sustained hypoxia [47]. Genes associated with energy metabolism and oxygen transmission were related to hypoxia adaptation of the Tibetan antelope [50]. Some types of cells adapt themselves well to hypoxia through enhanced glycolysis and down-regulation of oxidative phosphorylation [48]. Hypoxia can increase the production of reactive oxygen species and cause oxidative stress in animals [49]. The Tibet chicken had strong antioxidant stress ability to adapt to hypoxia through glutathione enzymes of detoxification [49], which was consistent with the results of GO and KEGG enrichments in this study. *EPAS1* was an important candidate gene for hypoxia adaptation in human and dog [27]. In the present study, we found *EPAS1* be a differentially expressed gene (Appendix A), though not detected by XPEHH or the Fst method, which might imply that *EPAS1* also plays a role in chicken embryonic hypxoia adaptation. 

A total of 353 DEKGs (Appendix A; Figure 3B) were found between the embryonic hearts of the Tibet chickens and the lowland chickens. The GO and KEGG analyses were also performed with these DEKGs and the results are displayed in Figure 5. From Figure 5, we can see that many terms were obviously enriched relating to sodium ion and amino acid transport (such as GO:0006814 ~ sodium ion transport; GO:0015190 ~ L-leucine transmembrane transporter activity; GO:0015191 ~ L-methionine transmembrane transporter activity; GO:0015808 ~ L-alanine transport; GO:0015810 ~ aspartate transport) and gene expression regulation (such as GO:0003682 ~ chromatin binding; GO:0045892 ~ negative regulation of transcription, DNA-templated; GO:0044822 ~ poly(A) RNA binding; GO:0000166 ~ nucleotide binding; GO:0007266 ~ Rho protein signal transduction; GO:0022627 ~ cytosolic small ribosomal subunit; GO:0005096 ~ GTPase activator activity; GO:0001047 ~ core promoter binding), cell adhesion (GO:0007155 ~ cell adhesion; gga04520: adherens junction; gga04514: cell adhesion molecules (CAMs)) and angiogenesis (GO:0001525 ~ angiogenesis), etc. These results were consistent with previous studies of different animal hearts in hypoxia [9,50,53,54,55,56]. Regulating angiogenesis may be one of the important approaches for animals to adapt to hypoxia [9,50,53]. Zhang et al. found only one credible candidate gene (*CTGF*) for chicken hypoxia adaptation, an important function of which was to regulate angiogenesis [9]. Some positive selection genes in highland American pika were enriched in regulation of angiogenesis [50] and hypoxia could also stimulate myocardial angiogenesis in rats via redox-regulated transcription factors [53]. Hypoxia could make the interventricular heterogeneity of rat hearts in the Na+ distribution more pronounced, the right ventricle was more prone to hypoxic damage, as it was less efficient in recruiting glucose as an alternative fuel and was particularly dependent on the efficient Na, K-ATPase function [54]. Adhesion is vital for the cellular structure maintenance. Hypoxia could impair the early adhesion of single lymphoma cells [55], which implies the role of cell adhesion in animal hypoxia adaptation. The results suggested that biological functions such as sodium ion and amino acid transports, gene expression regulation, cell adhesion and angiogenesis may play important roles in maintaining the normal function and development of chicken embryonic hearts under hypoxic conditions.

A total of 32 overlapping DEKGs (Appendix A; Figure 3B) were found between the DEKGs of hearts and livers. These genes may play important roles in both embryonic hearts and livers to fit the hypoxic circumstances. Reactive oxygen species (ROS) can be generated at mitochondrial complex III and stabilize HIF-1 in hypoxia [56]. Oxidative stress and angiogenesis were also closely related to hypoxia, as described above [49,50,53]. Among these 32 genes, at least six genes, including *ARNT* (aryl hydrocarbon receptor nuclear translocator, hypoxia-inducible factor 1-beta), *GADD45A* (growth arrest and DNA-damage-inducible 45 alpha), *IDH1* (isocitrate dehydrogenase 1 (NADP+)), *NRP1* (neuropilin 1), *ROMO1* (reactive oxygen species modulator 1) and *TFPI* (lipoprotein-associated coagulation inhibitor) could be regarded as important candidate genes for animal hypoxia adaptation since these genes were closely related to HIF-1, ROS, oxidative stress, angiogenesis and vascular repair [57,58,59,60,61,62,63,64]. ARNT is a constituent subunit of HIF-1 and hypoxia can inhibit induction of aryl hydrocarbon receptor activity in an ARNT-dependent manner [57]. ROMO1 is a novel protein which induces ROS production in the mitochondria [58] and is involved in diseases associated with oxidative stress [59]. IDH1 and GADD45A have also been related to oxidative stress [60,61,62]. IDH1 has a physiological function in protecting cells from oxidative stress by regulating the intracellular NADP+/NADPH ratio [60]. GADD45A was an important sensors of oxidative stress and could be induced by hypoxia [61], GADD45A knockdown could reduce oxidative stress by suppressing the p38 MAPK signaling pathway in CRL1730 human umbilical vein endothelial cells (HUVEC) [62]. TFPI played an inhibitory role on angiogenesis by regulating the proangiogenic effects of tissue factors or blocking VEGFR2 activation directly and decreasing endothelial cell migration through peptides within its carboxyl terminus [63]. Nrp1 is expressed in a variety of cells. Blocking Nrp1 can inhibit angiogenesis and impair survival and proliferation of multiple cancer cell types [64].

In the present study, a total of 327 differentially expressed unknown genes (DEUGs; Appendix A) were predicted and detected between embryos of the Tibet chicken and the lowland chicken, including 71 unknown genes differentially expressed in the liver, 271 unknown genes differentially expressed in the heart and 15 overlapping unknown genes that were differentially expressed in both organs. All these DEUGs might be related with chicken embryo hypoxia adaptation.

### 3.3. Overlapping Genes between the Results of Identification of Signatures of Selection and RNA Sequencing Analysis

Of all 705 DEKGs, 89 genes (Appendix A; Figure 3C), including 52 differentially expressed in liver, 40 differentially expressed in heart and 3 overlapping in both liver and heart, were also screened out by selection signature analyses. After analyses of GO and KEGG enrichments using the 52 differentially expressed genes in liver, we found 4 GO terms (GO:0004874 ~ aryl hydrocarbon receptor activity; GO:0034751 ~ aryl hydrocarbon receptor complex; GO:0014070 ~ response to organic cyclic compound; GO:0009410 ~ response to xenobiotic stimulus) were enriched with only two genes, *ARNT* and *AHR* (aryl hydrocarbon receptor). AHR and AHR/ARNT complexes were both transcription factors that controlled many genes’ expression [65]. HIF-1 is a heterodimer transcription factor consisting of two subunits, HIF-1α and ARNT (also named HIF-1β) [51,57,66]. Hypoxia can stimulate the expression of HIF-1α [67]. In comparison with lowland chickens, the embryonic brain of the Tibet chicken showed significantly higher expression levels of HIF-1α and the expression value of Tibet chicken embryo in hypoxia (14% oxygen content) was close to that of lowland chicken embryos incubated with normoxia [67]. ARNT was a common constituent subunit of the HIF-1 and AHR/ARNT complexes, and reciprocal crosstalk may occur [51,57,66]. Therefore, we speculated that the different expression levels of AHR and ARNT may affect the function of HIF-1 and thus affect the adaptation of chicken embryos to the hypoxic incubation environment. Hypoxia increased lactic acid content in blood, which could destroy the acid–base balance of animal body fluids [68], whereas gluconeogenesis could convert lactic acid into glucose [69]. It was reported that AHR could affect gluconeogenesis [70]. Therefore, we hypothesize that differential expression of AHR affected the lactic acid content of chicken embryo blood in a hypoxic environment. The 40 genes differentially expressed in the heart were enriched into three GO terms (GO:0046329 ~ negative regulation of JNK cascade; GO:0016477 ~ cell migration; GO:0001046 ~ core promoter sequence-specific DNA binding). JNK (c-Jun N-terminal protein kinase) is a class of stress-activated protein kinase, and its activation induced by hypoxia is an early response to hypoxic stress [71,72]. It was reported that JNK2 upregulated HIFs (Hypoxia Inducible Factors) and contributed to pulmonary hypertension and polycythemia induced by hypoxia [72]. Therefore, genes enriched for GO:0046329 (~ negative regulation of JNK cascade) in the present study (*ZMYND11* and *PAFAH1B1*) may play roles in the hypoxia adaptation of chicken embryos. The *ARNT* gene was also a differentially expressed gene in heart and enriched in the term GO:0001046 ~ core promoter sequence-specific DNA binding, which reflected the importance of the *ARNT* gene in the hypoxia adaptation of chicken embryos, whether in the heart or liver. 

Except for the several genes mentioned above, all these 89 genes detected by both methods (signatures of selection and RNA sequencing analysis) could be regarded as believable candidate genes for hypoxia adaptation in chicken embryos, and they may have effects on different aspects of chicken embryonic hypoxia adaptation. For example, GSTK1 (glutathione S-transferase kappa 1, also known as GST) is an important member of the glutathione antioxidant system, which plays an important role in the hypoxia adaptation of the Tibet chicken [49,73], and FGFR1 (fibroblast growth factor receptor 1) had an effect on the expression of HIF1α [74].

## 4. Conclusions 

In the present study, we performed selection signature and RNA sequencing analyses to detect candidate genes for embryonic hypoxia adaptation in the Tibet chicken. A total of 2287 candidate genes for the Tibet chicken’s hypoxia adaption were identified by XPEHH test or/and Fst test and 705 differentially expressed genes in embryonic heart or/and liver played important roles in the hypoxia adaptation of chicken embryos. Eighty-nine of believable candidate genes for embryonic hypoxia adaptation were screened out by both methods, namely, signatures of selection and RNA sequencing analysis. Some genes, such as *ARNT*, *AHR*, *GSTK1*, *FGFR1*, etc., could be regarded as the most important candidate genes for our future studies.

## Figures and Tables

**Figure 1 genes-11-00823-f001:**
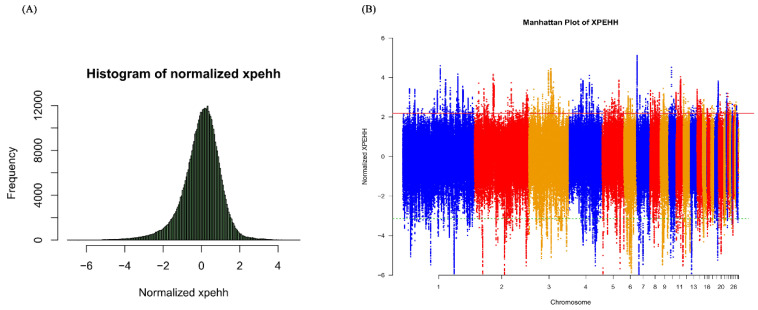
The distribution of cross population extended haplotype homozygosity (XPEHH) scores. (**A**) Histogram of normalized XPEHH; (**B**) XPEHH scores along autosomes. Unusually positive XPEHH scores suggest signatures of positive selection in the Tibet chicken genome. The red line shows the *p* value threshold of 0.01 (XPEHH score > 2.186052).

**Figure 2 genes-11-00823-f002:**
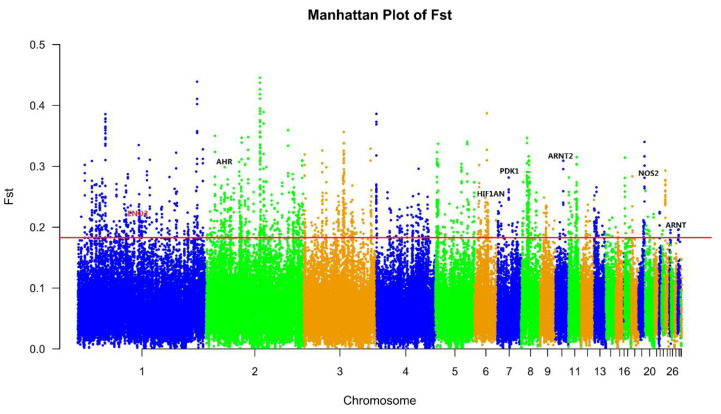
Values of fixation index (Fst) statistics along autosomes. Unusually positive Fst scores suggest signatures of positive selection in the Tibetan chicken and lowland chicken genome. The red line shows the p_value threshold of 0.02 (Fst > 0.1824514).

**Figure 3 genes-11-00823-f003:**
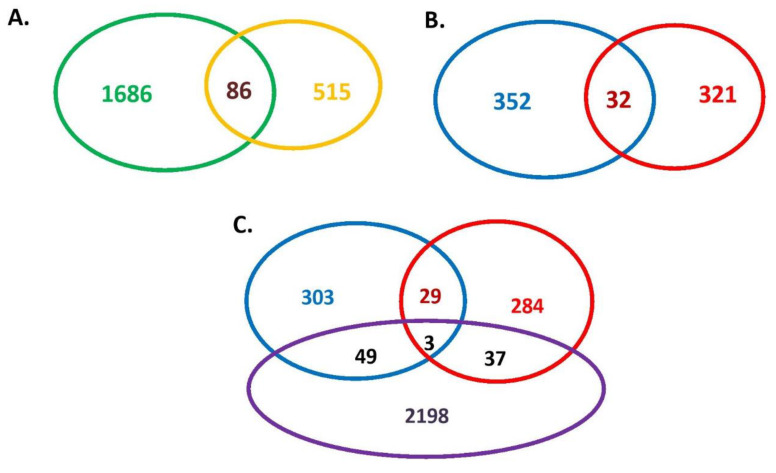
Numbers of genes detected with selection signature analyses and/or RNA sequencing analysis. (**A**) The green circle shows the number of genes detected by XPEHH (1686 + 86 = 1772) and the yellow circle shows the number of genes detected by the Fst method (515 + 86 = 601). The overlapping region shows the number of genes detected by both methods in common (86). The total number of genes checked out by selection signature analyses is 2287 (1772 + 601 − 86 = 2287). (**B**) The blue circle shows the number of differentially expressed known genes (DEKGs) in embryonic liver in a comparison of the Tibet chicken with the lowland chicken (352 + 32 = 384) and the red circle show the DEKG numbers in embryonic heart (321 + 32 = 353). The overlapping region shows the number of DEKGs both in embryonic liver and heart (32). (**C**) The genes of the blue circle ((303 + 49) + (29 + 3) = 352 + 32 = 384) and the red circle ((284 + 37) + (29 + 3) = 321 + 32 = 353) show the same genes as the Diagram. The purple circle shows the number of genes detected by XPEHH and/or Fst methods which is the same value as the gene number in Plot A (49 + 3 + 37 + 2198 = 2287). Of all 705 DEKGs (384 + 353 − 32 = 705), 89 genes including 52 differentially expressed in liver, 40 differentially expressed in heart and 3 overlapping in both of liver and heart, were also screened out by selection signature analyses. The 89 genes were considered important candidate genes for hypoxia adaptation in chicken embryos.

**Figure 4 genes-11-00823-f004:**
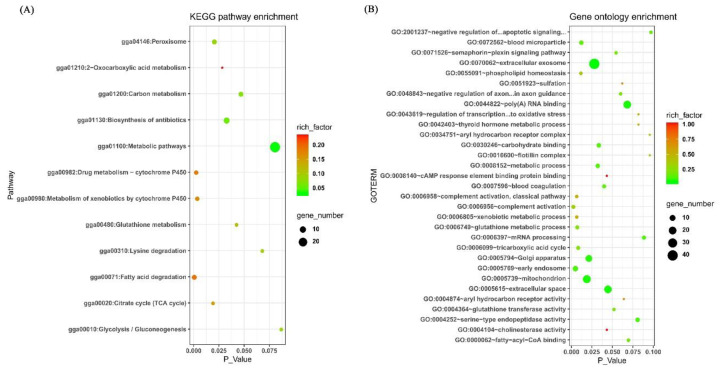
Kyoto Encyclopedia of Genes and Genomes (KEGG) and Gene Ontology (GO) analyses with differentially expressed genes between the livers of the Tibet chicken embryo and the lowland chicken embryo in a simulated hypoxic hatchibator with 13% oxygen concentration at the end of day 16 of incubation. (**A**) KEGG pathway analysis; (**B**) GO analysis.

**Figure 5 genes-11-00823-f005:**
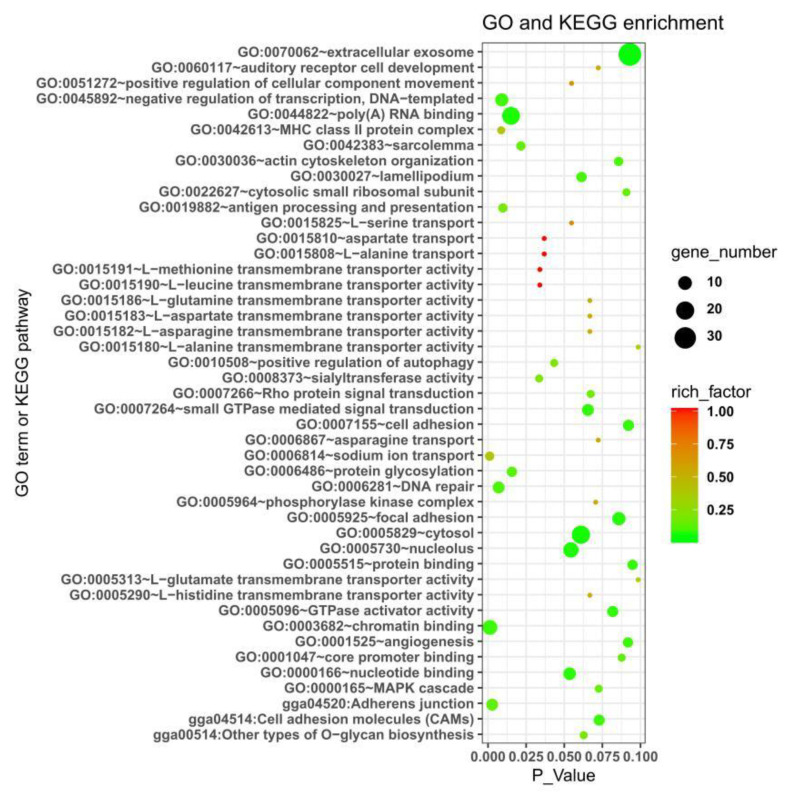
GO and KEGG enrichments with differentially expressed genes between the hearts of the Tibet chicken embryo and the lowland chicken embryo in a simulated hypoxic hatchibator with 13% oxygen concentration at the end of day 16 of incubation.

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
