# Peer review of "Identifying Candidate Genes for Hypoxia Adaptation of Tibet Chicken Embryos by Selection Signature Analyses and RNA Sequencing"

_genes, 2020, doi:10.3390/genes11070823_

Round 1

Reviewer 1 Report

In general, the work is interesting, the objectives of this study is to obtain candidate genes associated with hypoxia adaptation in the Tibet chicken embryos.
The authors indicate / line 56-57 / that some candidate genes for hypoxia adaptation /EPAS1, EGLN1, CTGF and AMOT/ had been detected out earlier. It was confirmed in own research?. Why 89 genes were considered as candidate genes for hypoxia adaptation in chicken embryos. In my opinion, these are genes that differ the Tibet chicken and lowland chicken /White leghorn, Silkie and Shouguang chicken /. They probably relate to many features that differentiate these breeds, and not just this one - hypoxia adaptation.

line 187.  The red dash line shows....;  This line is hardly visible.

Author Response

Responses for reviewers' comments

REVIEWER1

Comments and Suggestions for Authors

In general, the work is interesting, the objectives of this study is to obtain candidate genes associated with hypoxia adaptation in the Tibet chicken embryos.

  1. The authors indicate / line 56-57 / that some candidate genes for hypoxia adaptation /EPAS1, EGLN1, CTGF and AMOT/ had been detected out earlier. It was confirmed in own research?.

Response:

Not all of them. In our experiment, we found EPAS1 be a differentially expressed gene between the embryonic livers of the Tibet chicken and the white leghorn chicken (Table S4). Although we did not detect the AMOT gene, we detected AMOTL1 as candidate identified by XPEHH test(Table S1) and AMOTL2 as differentially expressed gene between the embryonic hearts of the Tibet chicken and the lowland chicken (Table S5). We did not detect out EGLN1 and CTGF genes in our research.

Line 211: Add sentences "EPAS1 was an important candidate gene for hypoxia adaptation in human and dog [27]. In the present study, we found EPAS1 be a differentially expressed gene (Table S4) but not detected out by XPEHH or Fst method, which might imply that EPAS1 also play a role in chicken embryonic hypxoia adaptation. " at the end of the paragraph.

  1. Why 89 genes were considered as candidate genes for hypoxia adaptation in chicken embryos. In my opinion, these are genes that differ the Tibet chicken and lowland chicken /White leghorn, Silkie and Shouguang chicken /. They probably relate to many features that differentiate these breeds, and not just this one - hypoxia adaptation.

Response: 

We detected the candidate genes with 3 steps as follows:

First, we performed two selection signature methods (FST and XPEHH) and detected out 2287 genes (FST:601+XPEHH:1772-Overlap:86=2287); Then, We analyzed the DEGs of RNA-Seq data of E16 embryos heart and liver from Tibetan chicken and White leghorn incubated at hypoxia and got 353 DEKGs from heart tissues and 384 DEKGs in liver. At last, We combined the results from genomic selection signature analyses and RNA sequencing analysis.

Genes which were screened out by selection signature analyses included genes related to hypoxia adaption and other features, and then we used RNA-Seq data from important tissues (embryonic heart and liver) under hypoxic condition to narrow the scope of candidate genes associated with hypoxia adaption. Our objective was to Identify Candidate Genes for Hypoxia Adaptation of Tibet Chicken and provide clues for our further studies to explain the molecular mechanism of hypoxia adaptation in the Tibet chicken embryos.

Line 23: add "hatched under hypoxia condition" after " lowland chicken embryos".

Line 68: add " under hypoxia condition" after "from embryonic livers and hearts".

  1. line 187.  The red dash line shows....;  This line is hardly visible.

Response:

Figure 1 and Figure 2: We replotted dash lines in Figure 1 and Figure 2.

Line166: change "The blue dash line" to "The red line".

Line183 (Line 187 in pdf version): change "The red dash line" to "The red line".

Reviewer 2 Report

Introduction

1/ Can you provide the scientific name of the Tibet Chicken and precise if it is a domestic or a wild species.

2/ L57: Please define HP method

3/ Please add a paragraph describing a bit animal adaptation to altitude and hypoxia.

4/ Can you discuss a bit more how signature of positive detections works and why this is hard to detect sometimes (Weigand and Leese, 2018, Zoological Journal of the linen society).

Methods

1/ L82-85: can you give a bit more detail about the type/Subspecies? of chicken? Why this one and why are they different?

2/ L123: Can you give more details about the DEG analysis. What are the quality score? Are the data normalized? Can you give more details about the softwares you used for the analysis?

Results/ Discussion

1/ L189-198: is it not better in the methods ? or a bit in the intro too, to explain to the non familiar reader what are this test in one/two sentence(s).

2/ L200-216: Please adjust the paragraph.

3/ L250- 277: Clarity and discussion should be really more improved. As i said before, it is hard to get an idea about the animal adaption to altitude and/or hypoxia at the gene level in your manuscript. You should more integrate your results in the existing literature rather than a long list not always very informative for the reader.

4/ L306-364: The clarity of this paragraph should be improved. Once again, be careful with the list effect. I suggest you to integrate more our interesting results in the relevant littérature and to tell more about them: Why are they important? Role in other known species? etc …. to get a clear view of the mechanisms playing in the chicken hypoxia adaptation.

Discussion of the present manuscript should be improved or integrated in an independent part. Also, the results should be « valorized » by integrating them in a better way to the existing literature of animal adaptation to hypoxia and altitude. Although, very little is described about chicken embryonic development and the interactions with the environment, and how that can explain the differences in hatching.

Author Response

Responses for reviewers' comments

REVIEWER2

Comments and Suggestions for Authors

Introduction

1/ Can you provide the scientific name of the Tibet Chicken and precise if it is a domestic or a wild species.

Response:

Line 18: Insert "(Gallus Gallus)" after "The Tibet Chicken".

Line 23: replace "the Tibet chicken and lowland chicken embryos" with " the Tibet chicken and White leghorn ( Gallus Gallus,a kind of lowland chicken) embryos".

Line 38: Insert "(Gallus Gallus, a domestic species)" between "chicken" and "originates".

2/ L57: Please define HP method

Response:

The pooled heterozygosity (Hp), described in detail by Rubin et al. (2010), is one of statistical methods to search signatures of positive selection with pooled next generation sequencing data. We have defined it in Line 52 (Line 53 in pdf version).

Rubin, C.J.; Zody, M.C.; Eriksson, J.; Meadows, J.R.; Sherwood, E.; Webster, M.T.; Jiang, L.; Ingman, M.; Sharpe, T.; Ka, S., et al. Whole-genome resequencing reveals loci under selection during chicken domestication. Nature 2010, 464, 587-591, doi:10.1038/nature08832.

3/ Please add a paragraph describing a bit animal adaptation to altitude and hypoxia.

Response:

Line 41: add following sentences "The White leghorn (Gallus Gallus), Shouguang chicken (Gallus Gallus) and Silkie chicken (Gallus Gallus) are all lowland domestic chicken breeds and originate from lowland areas with no more than 1000 m altitude. They do not adapt to high altitude hypoxia environment. In the lowland area (50 m, Beijing) , the Tibet chicken and lowland chickens show similar hatchabilities [1,7]; Under the low oxygen condition of 15% (equivalent to the partial pressure of O2 at a 2,900 m altitude in Linzhi, China), the hatchability of the Tibet chicken embryo could still reach about 80% and the hatching rate of lowland chicken was usually less than 40%[1,7]; Under the low oxygen condition of 13% (3650 m, Lhasa,China), the hatchability of the Tibet chicken was about 30%, while the hatchabilities of lowland chicken breeds were less than 10% [7]." at the end of the paragraph.

4/ Can you discuss a bit more how signature of positive detections works and why this is hard to detect sometimes (Weigand and Leese, 2018, Zoological Journal of the linen society).

Response

Line 43: Insert "with hard selective sweeps were easier to be detected out than those with soft selective sweeps and" between "regions" and "at least".

Line 44: change "[13-15]" to "[13-16]".

Line 49: Insert sentences"Soft selective sweeps are more common and also more difficult to detect out than hard sweeps because of their weaker effects on linked sites [16,17]. Some confounding factors, such as purifying and background selection, demography and Migration can affect the results of positive selection analysis [16]."before "Many statistical methods".

Line 431: Insert "16.  Weigand, H.; Leese, F. Detecting signatures of positive selection in non-model species using genomic data. Zoological Journal of the Linnean Society 2018, 184, 528-583, doi:10.1093/zoolinnean/zly007.

  1. Pritchard, J.K.; Joseph K. Pickrell, J.K.; Coop, G. The genetics of human adaptation: hard sweeps, soft sweeps, and polygenic adaptation. Current Biology2010, 20, R208-R215, doi:10.1016/j.cub.2009.11.055.".

Reorder references.

Methods

1/ L82-85: can you give a bit more detail about the type/Subspecies? of chicken? Why this one and why are they different?

Response:

Line41: Insert the following sentences "The White leghorn (Gallus Gallus), Shouguang chicken (Gallus Gallus) and Silkie chicken (Gallus Gallus) are all lowland domestic chicken breeds and originate from lowland areas with no more than 1000 m altitude. They do not adapt to high altitude hypoxia environment. In the lowland area (50 m, Beijing) , the Tibet chicken and lowland chickens show similar hatchabilities [1,7]; Under the low oxygen condition of 15% (equivalent to the partial pressure of O2 at a 2,900 m altitude in Linzhi, China), the hatchability of the Tibet chicken embryo could still reach about 80% and the hatching rate of lowland chicken was usually less than 40%[1,7]; Under the low oxygen condition of 13% (3650 m, Lhasa, China), the hatchability of the Tibet chicken was about 30%, while the hatchabilities of lowland chicken breeds were less than 10% [7]." at the end of Line 41.

2/ L123: Can you give more details about the DEG analysis. What are the quality score? Are the data normalized? Can you give more details about the softwares you used for the analysis?

Response:

Line 99: replace "After quality control of sequencing raw data by the commercial service companies, clean data were " with "After the adapter sequences in raw data trimmed, reads with more than 10% of undetermined bases (N) were removed, clean reads were obtained after removing low-quality bases (Qphred < 20), and then".

Line 121: add the sentence "Expression level for each gene was estimated with the FPKM (Fragments per kilobase of transcript per million fragments mapped) value. " at the front of the paragraph.

Line 195: delete the sentence "Expression level for each gene was estimated with the FPKM (Fragments per kilobase of transcript per million fragments mapped) value. ".

Line 123-125: replace "Transcripts were assembled, quantified and merged with StringTie (Version 2.0; [42]). Differentially expressed transcripts and genes were screened with Ballgown (Version 2.12.0; [43])." with " The SAM files were sorted and converted to BAM files with SAMtools (https://github.com/samtools/samtools/releases/tag/1.9, Version 1.9). Transcripts of each sample were assembled with StringTie (Version 2.0; [42]) based on Gallus_gallus.GRCg6a.97.gtf (ftp://ftp.ensembl.org/pub/release-97/gtf/gallus_gallus/) and each sample's BAM file. After assemblies, samples' gtf documents and Gallus_gallus.GRCg6a.97.gtf were merged together with stringtie --merge option and transcript abundances of each sample were estimated with stringtie -eB option [41]. Transcripts with a variance across samples less than one were removed and then differentially expressed transcripts and genes between embryos of the Tibet chicken and White leghorn were screened using the stattest function from Ballgown (Version 2.12.0; [43]) with the getFC=TRUE parameter.".

Results/ Discussion

1/ L189-198: is it not better in the methods ? or a bit in the intro too, to explain to the non familiar reader what are this test in one/two sentence(s).

Response:

Line189-191: replace "Although the Fst method is also sensitive in detecting fixed selection signatures [12], the correlation between the results of the two methods was very low (0.03) in chicken data [45]. This may indicate that the two methods have different priorities in detecting the selected genes." with "Although the Fst method is also sensitive in detecting fixed selection signatures [12], the test power of XPEHH is significantly higher than that of Fst method under the same parameters of marker interval distance, frequency of the selected allele, sample size or selection coefficient [45]. The correlation between the results of the two methods was very low (0.03) in chicken data [45]. In the present study, less than 5% of the 1771 genes were also detected by Fst, and most of the genes detected by Fst (85.6%) were not detected by XPEHH, which may indicate that the two methods have different priorities in detecting the selected genes. The overlapping genes may indeed be related to positive selection signatures, but we can not rule out others.".

2/ L200-216: Please adjust the paragraph.

Response:

Line195-211 (Line200-216 in pdf version) : The format of the paragraph has been adjusted.

3/ L250- 277: Clarity and discussion should be really more improved. As i said before, it is hard to get an idea about the animal adaption to altitude and/or hypoxia at the gene level in your manuscript. You should more integrate your results in the existing literature rather than a long list not always very informative for the reader.

Response:

Line258 (Line263 in pdf version): Insert sentences "Regulating angiogenesis may be one of the important approaches for animals to adapt hypoxia [9,50,57]. Zhang et al. found only one credible candidate gene (CTGF) for chicken hypoxia adaptation, an important function of which was to regulate angiogenesis [9]. Some positive selection genes in highland American pika were enriched in regulation of angiogenesis [50] and hypoxia could also stimulate myocardial angiogenesis in rat via redox-regulated transcription factor [57]. Hypoxia could make the interventricular heterogeneity of rat hearts in the Na+ distribution more pronounced, the right ventricle was more prone to hypoxic damage, as it was less efficient in recruiting glucose as an alternative fuel and was particularly dependent on the efficient Na, K-ATPase function [52]. Adhesion is vital for the cellular structure maintenance, hypoxia could impaired the early adhesion of single lymphoma Cell [53], which implied the role of cell adhesion in animal hypoxia adaptation." before "The results suggested that".

Line266-267 (Line269-270 in pdf version): replace "Reactive oxygen species, oxidative stress and angiogenesis were closely related to hypoxia adaptation [28,60,61]." with "Reactive oxygen species (ROS) can generated at mitochondrial complex III and Stabilize HIF-1 in hypoxia [60]. Oxidative stress and angiogenesis were also closely related to hypoxia as described above [28,49,50,57].".

Line271-273 (Line275-277 in pdf version):replace "since ARNT was a constituent subunit of hypoxia-inducible factor 1 and the other gene functions were related to reactive oxygen species, oxidative stress, angiogenesis and vascular repair [62-69]." with "since these genes were colsely related to HIF-1, ROS, oxidative stress, angiogenesis and vascular repair [62-69]. ARNT was a constituent subunit of HIF-1 and hypoxia can inhibit induction of aryl hydrocarbon receptor activity on an ARNT-dependent manner[63]. ROMO1 was a novel protein which induced ROS production in the mitochondria [62] and involved in diseases associated with oxidative stress [68]. IDH1 and GADD45A were also related to oxidative stress[69,66,67]. IDH1 had a physiological function in protecting cells from oxidative stress by regulating the intracellular NADP+/NADPH ratio[69]. GADD45A was an important sensors of oxidative stress and could be induced by hypoxia [66], GADD45A knockdown could reduce oxidative stress by suppressing the p38 MAPK signaling pathway in CRL1730 human umbilical vein endothelial cell (HUVEC)[67]. TFPI played an inhibitory role on angiogenesis by regulateing the proangiogenic effects of tissue factors, or blocking VEGFR2 activation directly and decreasing endothelial cell migrationin through peptides within its carboxyl terminus [64]. Nrp1 is expressed in a variety of cells. Blocking Nrp1 can inhibit angiogenesis and impair survival and proliferation of multiple cancer cell types [65].".

4/ L306-364: The clarity of this paragraph should be improved. Once again, be careful with the list effect. I suggest you to integrate more our interesting results in the relevant littérature and to tell more about them: Why are they important? Role in other known species? etc …. to get a clear view of the mechanisms playing in the chicken hypoxia adaptation.

Response:

Line317 (Line 315 in pdf version). Insert "Hypoxia can stimulate the expression of HIF-1α [72+]. In comparison with lowland chicken, the embryonic brain of the Tibet chicken showed significantly higher expression level of HIF-1α and the expression value of Tibet chicken embryo in hypoxia (14% oxygen content) was close to that of lowland chicken embryos incubated with normoxia [72+]." before "ARNT was a common".

Line 583 (Line586 in pdf version): Insert a reference "72. Wang, C.F., Wu, C.X.; Li, N. Differential gene expression of hypoxia inducible factor-1alpha and hypoxic adaptation in chicken. Yi Chuan 2007, 29(1), 75-80, doi: 10.1360/yc-007-0075.".

Line 271-273: we have already inserted a sentence of "ARNT was a constituent subunit of HIF-1 and hypoxia can inhibit induction of aryl hydrocarbon receptor activity on an ARNT-dependent manner[63]."

Discussion of the present manuscript should be improved or integrated in an independent part. Also, the results should be « valorized » by integrating them in a better way to the existing literature of animal adaptation to hypoxia and altitude. Although, very little is described about chicken embryonic development and the interactions with the environment, and how that can explain the differences in hatching.

Response:

Line211 (Line216 in pdf version): change "[45]" to "[45-50]"; Add sentences "Hypoxia-Inducible Factor 1 (HIF-1) was one of the key transcription factors for animal hypoxia adaptation [58,59]. Elevated fatty acid metabolism could decrease the stabilization of the alpha subunit of HIF-1 which was vital for animal survival in hypoxia [46]. Skeletal muscle decreased fatty acid oxidation in sustained hypoxia [47]. Genes associated with energy metabolism and oxygen transmission were related to hypoxia adaptation of the Tibetan antelope [50+]. Some types of cells adapt themselves well to hypoxia through enhanced glycolysis and down-regulation of oxidative phosphorylation [48]. Hypoxia can increase the production of reactive oxygen species and cause oxidative stress in animals [49]. The Tibet chicken had strong antioxidant stress ability to adapt to hypoxia through glutathione enzymes of detoxification [49], which was consistent with the results of GO and KEGG enrichments in this study." at the end of the paragraph.

Line511 (Line513 in pdf version): Insert refrences "47. Dodd, M.S.; Fialho, M.L.S.; Aparicio, C.N.M.; Kerr, M.; Timm, K.N.; Griffin, J.L.; Luiken, J.J.F.P.; Glatz,  J.F.C.; Tyler, D.J.; Heather, L.C. Fatty acids prevent Hypoxia-Inducible Factor-1α signaling through decreased succinate in Diabetes. JACC Basic Transl. Sci. 2018, 3(4), 485-498, doi: 10.1016/j.jacbts.2018.04.005.

  1. Morash,A.J.; Kotwica,A.O.; Murray, A.J. Tissue-specific changes in fatty acid oxidation in hypoxic heart and skeletal muscle. Am. J. Physiol. Regul. Integr. Comp. Physiol. 2013, 305(5), R534-R541, doi: 10.1152/ajpregu.00510.2012.
  2. Malthankar-Phatak,G.H.; Patel,A.B.; Xia, Y.; Hong, S.; Chowdhury, G.M.I.; Behar, K.L.; Orina, I.A.; Lai, J.C.K. Effects of Continuous Hypoxia on Energy Metabolism in Cultured Cerebro-Cortical Neurons. Brain Res. 2008, 1229, 147-154, doi: 10.1016/j.brainres.2008.06.074.".

Line 41: Insert sentences " In the lowland area (50 m, Beijing) , the Tibet chicken and lowland chickens show similar hatchabilities [1,7]; Under the low oxygen condition of 15% (equivalent to the partial pressure of O2 at a 2,900 m altitude in Linzhi, China), the hatchability of the Tibet chicken embryo could still reach about 80% and the hatching rate of lowland chicken was usually less than 40%[1,7]; Under the low oxygen condition of 13% (3650 m, Lhasa, China), the hatchability of the Tibet chicken was about 30%, while the hatchabilities of lowland chicken breeds were less than 10% [7]." at the end of Line 41.

Others:

Line8: change "[email protected]" to "[email protected]".

Line 41: change "[8,9]" to "[1,7-9]".

Line 191: delete "selected".

References: Remove unused references and reorder them.

References: Adjust references' formats.

Round 2

Reviewer 2 Report

Dear authors, 

Thanks for the revisions. However, a point by point response letter can give to the reviewer a more easy task. Also, while i found the modifications positive, i suggest you to send a manuscript with a correct editing format. Indeed, some places in the text were not very easy to read and check. 

Also, i don t really understand why did you FPKM ? In general, DEG pipeline is enough unless you want to insist on specific transcripts and show them through a graphical representation that i did not find. Furthermore, while calculating the FPKM, a reference should be add to the text concerning the methods. 

Here are few comments: 

L136-137: missing citation for FPKM calculation. Why do you use FPKM calculation? I did not find any illustrations for that. Also, DE pipelines are enough to detect differentially expressed genes.

L189-192: Don t see anything on the legend, please clarify.

L255-262; Please put properly the figure under the text and not above.

While, you did really nice progress in the modification, i suggest to the editor Minor revisions before acceptance in order to get a manuscript in a proper format (maybe my computer i dont know because some figures are simply on the text) but also to avoid in some places the "list effect" with a discussion a bit more constructed and a better conclusion for future researches and possibilities. In the final version, please keep the modifications in red, that is very good, HOWEVER, please, accept the modifications to get a manuscript in a better format. 

Best regards. 

Author Response

L136-137: missing citation for FPKM calculation. Why do you use FPKM calculation? I did not find any illustrations for that. Also, DE pipelines are enough to detect differentially expressed genes.

Response: 

Line137: Insert"[40]" after "transcript per million fragments mapped) value".

Line547: Insert a reference "Trapnell, C.;  Williams, B.A.; Pertea, G.; Mortazavi, A.; Kwan, G.; Baren, M.J.v.; Salzberg, S.L.; Wold, B.J.; Pachter, L. Transcript assembly and abundance estimation from RNA-Seq reveals thousands of new transcripts and switching among isoforms. Nat Biotechnol. 2010, 28(5), 511-515, doi:10.1038/nbt.1621."    

Reorder references.

L189-192: Don t see anything on the legend, please clarify. 

Response:

Line189-192: Insert several important genes' names related to hypoxia adaptation (ENO2, AHR, HIF1AN,PDK1,ARNT2,NOS2,ARNT) in Figure 2.

L255-262; Please put properly the figure under the text and not above.

Response:

Move the Figure 3 from Line242-276 to Line 325.

While, you did really nice progress in the modification, i suggest to the editor Minor revisions before acceptance in order to get a manuscript in a proper format (maybe my computer i dont know because some figures are simply on the text) but also to avoid in some places the "list effect" with a discussion a bit more constructed and a better conclusion for future researches and possibilities. In the final version, please keep the modifications in red, that is very good, HOWEVER, please, accept the modifications to get a manuscript in a better format.

Response:

We corrected our manuscript's format according to your suggestion and the journal's guideline.